# Insights into substrate binding and utilization by hyaluronan synthase

**Zachery Stephens[1], Julia Karasinska[1†], Jochen Zimmer[1,2]***

[1]University of Virginia School of Medicine, Charlottesville, United States; [2]Howard Hughes Medical Institute, Chevy Chase, United States

## eLife Assessment

This study addresses a **fundamental** question in glycobiology by elucidating how a single-site processive enzyme orchestrates the alternating addition of sugars to synthesize complex polysaccharides such as hyaluronan. The findings are **compelling**, providing a clear mechanistic framework supported by strong experimental validation. Major strengths include the integration of high-resolution structural data with rigorous biochemical analyses, resulting in a well-supported model of hyaluronan assembly.

**\*For correspondence:**
jz3x@virginia.edu

**Present address:** [†]Institute of Molecular Biology and Biophysics, ETH Zurich, HPK, Zurich, Switzerland

**Competing interest:** The authors declare that no competing interests exist.

## Abstract

Hyaluronan (HA), a heteropolysaccharide of alternating *N*-acetylglucosamine (GlcNAc) and glucuronic acid (GlcA), is an essential component of the vertebrate extracellular matrix. HA biosynthesis proceeds via three evolutionarily convergent reaction mechanisms, catalyzed by the membrane-integrated class 1 enzymes that either elongate the non-reducing (NR) or reducing end of HA, as well as the class 2 hyaluronan synthase (HAS), a soluble non-processive enzyme. Class 1-NR HAS, found in both vertebrates and large double-stranded DNA viruses, is monomeric and couples HA polymerization via coordinated transfer of UDP-GlcNAc and UDP-GlcA substrates with the secretion of the nascent HA chain through its own transmembrane channel. How this HAS discriminates between two UDP-sugars using a single active site is a critical, yet unresolved question. Using single-particle cryo-EM, we reveal a two-step process by which the *Chlorella* virus HAS (CvHAS) recognizes and positions its substrate, UDP-GlcA, for glycosyl transfer. Further, we report greatly diminished turnover of UDP-GlcA in the absence of a primer, distinguishing acceptor-free activity toward UDP-GlcNAc. Lastly, prompted by observation of a dodecyl maltoside bound HAS, we demonstrate the utility of non-canonical acceptors in priming of a UDP-GlcA transfer reaction. Altogether, this work clarifies the molecular basis for HAS' dual substrate specificity and the role of UDP-GlcA recognition in integrity of HA synthesis.

## Introduction

Hyaluronan (HA), a heteropolysaccharide of alternating glucuronic acid (GlcA) and *N*-acetylglucosamine (GlcNAc), is an abundant and essential extracellular matrix material in vertebrates. It performs a plethora of developmental and physiological functions with critical contributions to fertilization, cardio- and angiogenesis, wound healing, and joint lubrication (*Girish and Kemparaju, 2007*; *Simpson et al., 2022*; *Knudson et al., 2019*).

Biosynthesis of HA is catalyzed in a cation-dependent polymerization reaction with nucleotide-sugar substrates, UDP-GlcNAc and UDP-GlcA, by HA synthase (HAS) (*Weigel and DeAngelis, 2007*). Several unique mechanisms of HA synthesis have emerged, lending to the definition of processive class 1 HASs, transmembrane proteins which couple HA synthesis and transport activities, and distributive class 2 HASs, peripheral membrane proteins which lack a secretion function (*DeAngelis and*

*Zimmer, 2023*). HA polymerization by class 1 HAS can occur at either the non-reducing end (class 1-NR) as reported for the *Paramecium bursaria Chlorella virus* HAS, or the reducing end (class 1-R) in the case of the dimeric *S. equismillis* HAS (*Blackburn et al., 2018*). Class 1-NR HASs function as monomers and contain a single catalytic GT-A domain (*Taujale et al., 2020*), necessarily dictating that a single active site be sufficient to sequentially recognize and transfer both UDP-GlcA and UDP-GlcNAc to a nascent HA chain (*Maloney et al., 2022*).

Recent structural and biochemical analyses of Chlorella virus and *Xenopus laevis* isoform-1 HAS (CvHAS and XlHAS-1, respectively) provided detailed insights into the multitasking of type 1-NR HAS (*Maloney et al., 2022*; *Górniak et al., 2025*). First, to initiate HA biosynthesis, HAS binds and hydrolyzes UDP-GlcNAc, such that the released GlcNAc monosaccharide can prime polymer biosynthesis. Second, the GlcNAc-primed enzyme binds the next substrate, UDP-GlcA, leading to the formation of a ternary complex that facilitates glycosyl transfer and the formation of a β-linked GlcA(1,3)GlcNAc disaccharide(1-3) . The subsequent binding of a UDP-GlcNAc substrate molecule presumably translocates the HA disaccharide by an unknown mechanism and facilitates the transfer of GlcNAc to the nascent HA polymer, thereby forming a β-1,4 linkage to GlcA. These steps must be repeated thousands of times to synthesize and secrete an HA polysaccharide of greater than 10,000 disaccharide units (*Zhao et al., 2023*).

While UDP-GlcNAc is a common metabolite in all reported HA-producing species, the physiological concentration of UDP-GlcA, however, may limit HA biosynthesis under certain conditions (*Oikari et al., 2018*). Accordingly, the Chlorella virus encodes a UDP-glucose dehydrogenase enzyme that generates UDP-GlcA from UDP-glucose (*Van Etten et al., 2017*). Further, expressing this enzyme in engineered HA-producing systems increased the overall production levels (*Yu and Stephanopoulos, 2008*; *Cheng et al., 2016*).

In this study, we delineate differences in UDP-GlcA coordination by CvHAS using cryo-electron microscopy (cryo-EM). Enzymology and biochemical analysis were utilized to understand how UDP-GlcA interaction strength and turnover efficiency vary in the presence and the absence of a GlcNAc primer. We further show that CvHAS exhibits a degree of promiscuity toward acceptors for glycosyl transfer, enabling the biosynthesis of unnatural complex carbohydrates. Lastly, our structural analysis reveals a dodecyl maltoside-inhibited state of CvHAS in which the detergent molecule occupies the acceptor-binding site, thereby providing insights into HAS substrate promiscuity.

## Results

We used cryo-EM analysis to gain structural insights into the interaction of CvHAS with its substrate UDP-GlcA. The catalytically inactive (D302N) CvHAS mutant was reconstituted into MSPE3D1 lipid nanodiscs as described before (*Maloney et al., 2022*) and complexed with high affinity nanobodies that recognize cytosolic and extracellular epitopes. This complex was then incubated with 10 mM $MnCl_2$ and 5 mM UDP-GlcA in the absence of a receiving GlcNAc monosaccharide, prior to cryo grid preparation. The obtained cryo-EM dataset was processed as outlined in *Figure 1—figure supplement 1*.

### CvHAS binds UDP-GlcA in two different binding poses

Focused refinement of particles harboring a UDP-GlcA molecule at the active site resolved two substrate-binding poses, interpreted as proofreading and inserted states. The inserted pose resembles the previously reported UDP-GlcA conformation in the presence of a priming GlcNAc sugar (*Górniak et al., 2025*). In this state, the nucleotide's uracil moiety is surrounded by Tyr91, His174, and Lys177, its diphosphate group, together with CvHAS' DxD motif (Asp201 and Asp203), coordinates a divalent cation, most likely $Mn^{2+}$, and the glucuronic acid donor sugar resides in a pocket underneath the acceptor-binding site (*Figure 1B*). GlcA's ring oxygen resides within hydrogen bonding distance to the Nε of Trp342, and its C6 carboxylate group, although poorly resolved, is positioned near the guanidinium group of Arg341 (*Figure 1C, D*). The side chain's Nε is within 3.5 Å of GlcA's carboxyl group, potentially accounting for a weak electrostatic interaction. An alternative side chain conformation for Arg341 was well-supported by cryo-EM density and thus modeled. In this position, the residue's guanidinium group is about 4.6 Å from GlcA's C6 carboxylate (*Figure 1D*) and likely interacts with it

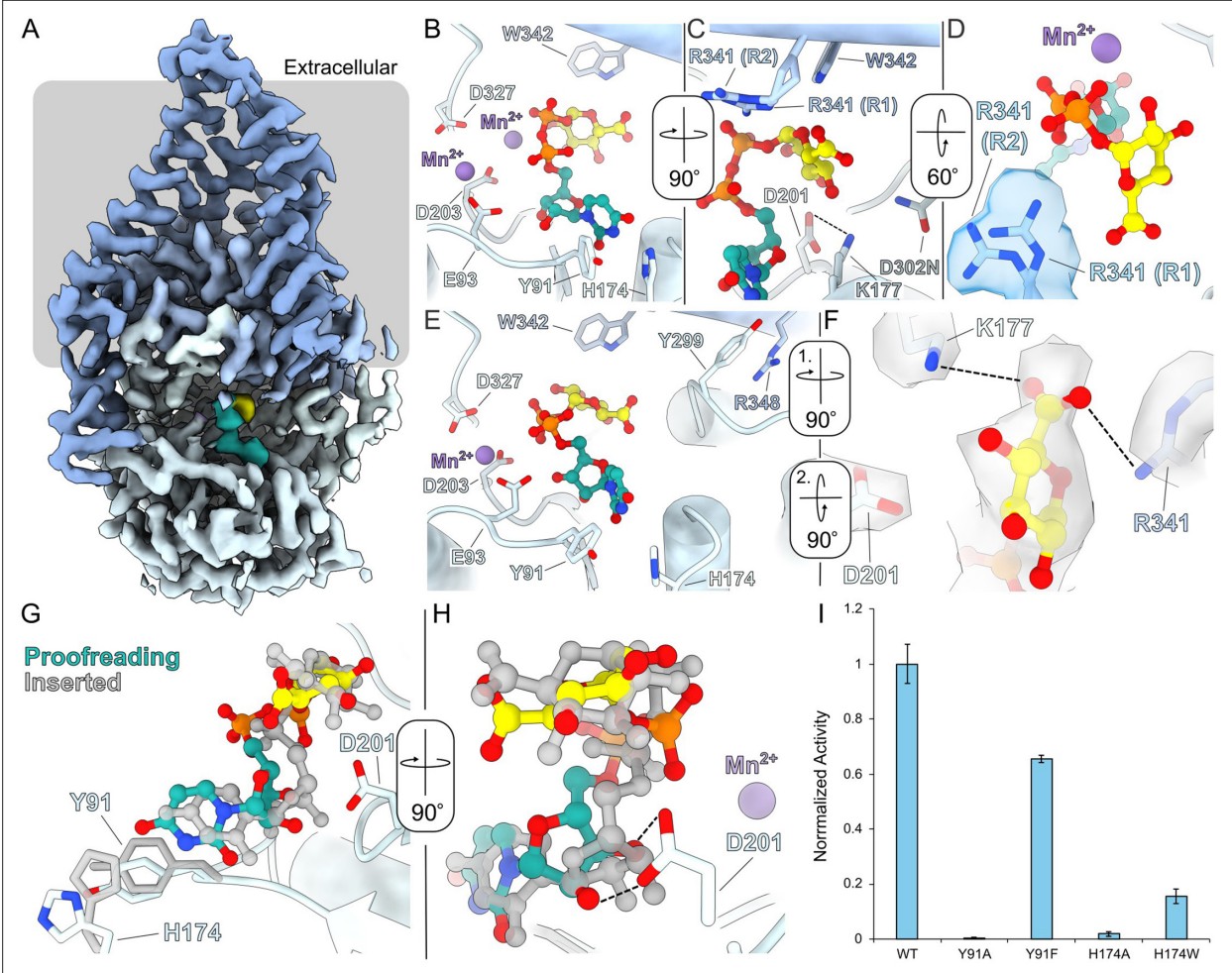

**Figure 1.** UDP-GlcA coordination and proofreading. (**A**) Cryo-EM density map of CvHAS bound to UDP-GlcA overlayed on a gray bar representing the membrane boundary. UDP-GlcA density is shown in light sea green and yellow for its UDP and GlcA moieties, respectively. (**B, C**) Coordination of an inserted UDP-GlcA substrate in the absence of a GlcNAc primer. UDP-GlcA is shown as a ball and stick model with light sea green carbon atoms for the UDP moiety and yellow carbon atoms for GlcA. (**D**) Cryo-EM density for Arg341 rotamer 1 (R1) and rotamer 2 (R2). (**E**) Coordination of UDP-GlcA in the proofreading conformation. (**F**) Cryo-EM density for UDP-GlcA's sugar ring and surrounding residues in the proofreading state. (**G, H**) Alignment of proofreading and inserted UDP-GlcA positions. The ligand in the inserted UDP-GlcA structure is shown in gray. (**H**) Activity of uracil-binding pocket mutants for CvHAS. Activity measurements were normalized to wild-type (WT) and are reported as the average of three technical replicates. Error bars represent the standard deviation from the mean.

The online version of this article includes the following source data and figure supplement(s) for figure 1:

**Figure supplement 1.** Cryo-EM data processing for UDP-GlcA inserted and proofreading structures.

**Figure supplement 2.** Cryo-EM density for Mn and the priming loop c-terminus in inserted and proofreading conformations.

**Figure supplement 3.** Purification CvHAS' uracil binding pocket mutants.

**Figure supplement 3—source data 1.** Raw image file for Coomassie-stained gel of CvHAS uracil-binding mutants and XlHAS-1.

**Figure supplement 3—source data 2.** Coomassie-stained gel of CvHAS uracil-binding pocket mutants and XlHAS-1 with relevant lanes labeled.

**Figure supplement 4.** Comparison of UDP-GlcA binding by CvHAS to UDP-GlcNAc binding by CHS.

via a mediating water molecule. In addition, the conserved Asp201 forms a salt bridge with Lys177 at the back of the nucleotide-binding pocket (*Figure 1C*).

Previous analysis identified a second divalent cation-binding site in CvHAS' catalytic pocket (*Maloney et al., 2022*). This site is created by Glu93 together with the C-terminal Asp of the DxD motif (Asp203) and appears to be metal occupied in nucleotide-bound and apo states. This is also the case in both of our new cryo-EM maps, suggesting that the site is sufficient to coordinate a hydrated

cation (*Figure 1—figure supplement 2C, D*). Further, this coordination site is required for catalytic activity as replacing Glu93 with Ala renders CvHAS inactive (*Maloney et al., 2022*).

In the second 'proofreading' UDP-GlcA-binding pose resolved in our dataset, the ligand's nucleotide and donor sugar moieties are less deeply inserted into the catalytic pocket (*Figure 1E–H*). The uracil moiety is shifted by about 1.5 Å toward the entrance of the catalytic pocket, resulting in the rotation of His174 away from Tyr91. Further, the nucleotide's ribose moiety is tilted away from the DxD motif, which places the attached diphosphate group also closer to the binding cleft's entrance. In this position, the diphosphate is suitably positioned to contribute to cation coordination at the second metal-binding site, together with Asp203 and Glu93 (*Figure 1E*). Indeed, we observe strong cryo-EM density at this site, consistent with an octahedrally coordinated manganese cation (*Figure 1—figure supplement 2D*).

The partial insertion of UDP-GlcA in the proofreading pose further allows Asp201 of the DxD motif to move away from its binding partner Lys177 to interact with the ribose's C2 and C3 hydroxyl groups (*Figure 1H*). The ammonium group of Lys177, instead, moves toward the donor sugar and, together with the guanidinium group of Arg341, sandwiches GlcA's carboxylate group (*Figure 1F*). Both side chains are well resolved in the cryo-EM map and about 3.5 Å away from the carboxylate group, which is also resolved at a slightly lower contour level (*Figure 1F*). Compared to the inserted UDP-GlcA state described above, these interactions create a basic pocket that recognizes GlcA's carboxylate group.

Next to the interactions with Lys177 and Arg341, GlcA's carboxylate is further framed by the C-terminal segment of the priming loop that leads into the conserved GDD motif (residues 300–302). The backbone conformation of the priming loop differs slightly in both UDP-GlcA-bound conformations. Most notably, while the entire backbone is well resolved in the inserted UDP-GlcA-bound pose (*Figure 1—figure supplement 2E*), the density of the conserved Gly300 is essentially absent in the proofreading state (*Figure 1—figure supplement 2F*), suggesting that this residue and parts of the preceding priming loop are flexible until UDP-GlcA is fully inserted into the catalytic pocket. The conserved Arg348 that belongs to an amphipathic helix at the cytosolic water–lipid interface interacts with the backbone carbonyl oxygen of Gly300 in the inserted state. When the priming loop is flexible in the proofreading state, however, Arg348 bends away by about 2 Å to adopt a cation-π stacking interaction with Tyr299 (*Figure 1E*). Arginine 348 is conserved among bacterial and eukaryotic HASs, and Tyr299 is conservatively substituted with Phe in vertebrate HASs, suggesting this interaction is preserved across species.

We next thought to validate the contribution of conformational changes in the uracil-binding groove to catalytic activity of CvHAS. To this end, we tested the functional relevance of Tyr91 and His174 through site-directed mutagenesis. CvHAS' activity can be quantified in vitro by measuring the accumulation of tritiated HA by scintillation counting, as previously described (*Blackburn et al., 2018*). Accordingly, replacing Tyr91 with Ala abolishes catalytic activity of CvHAS, while substituting the residue with Phe maintains about 80% activity, relative to the wild-type enzyme (*Figure 1I*). Similarly, drastic effects are observed when replacing His174 with either Ala or Trp, its corresponding substitution in vertebrate HASs. The Ala mutant is essentially inactive, while the H174W mutant retains about 25% of wild-type activity (*Figure 1I*). All mutants share similar size exclusion chromatography profiles with the WT enzyme, suggesting that the substitutions do not cause a folding defect (*Figure 1—figure supplement 3A–E*).

## CvHAS binds UDP-GlcA with low micromolar affinity

We performed isothermal titration calorimetry to determine the apparent affinity of UDP-GlcA binding to CvHAS. Titrating UDP-GlcA into a cell containing the catalytically inactive D302N CvHAS mutant at 45 µM revealed saturable exothermic heat responses that could be fit to a single binding isotherm. By this method, we obtained dissociation constants (Kd) for UDP-GlcA of 69 and 24 µM in the absence and the presence of a GlcNAc primer, respectively (*Figure 2A, B*). Under similar conditions, no binding was observed for UDP-GlcNAc, UDP alone, as well as UDP-glucose (*Figure 2—figure supplement 1*). While the reason for observing unfittable heat signatures when titrating UDP-GlcNAc is unknown, the absence of a thermal response during UDP and UDP-glucose titration points to the contribution of UDP-GlcA's carboxylate to binding.

We next thought to compare substrate turnover efficiency for UDP-GlcA and UDP-GlcNAc under initiation and synthesis conditions. To this end, an enzyme-coupled assay was utilized that links the

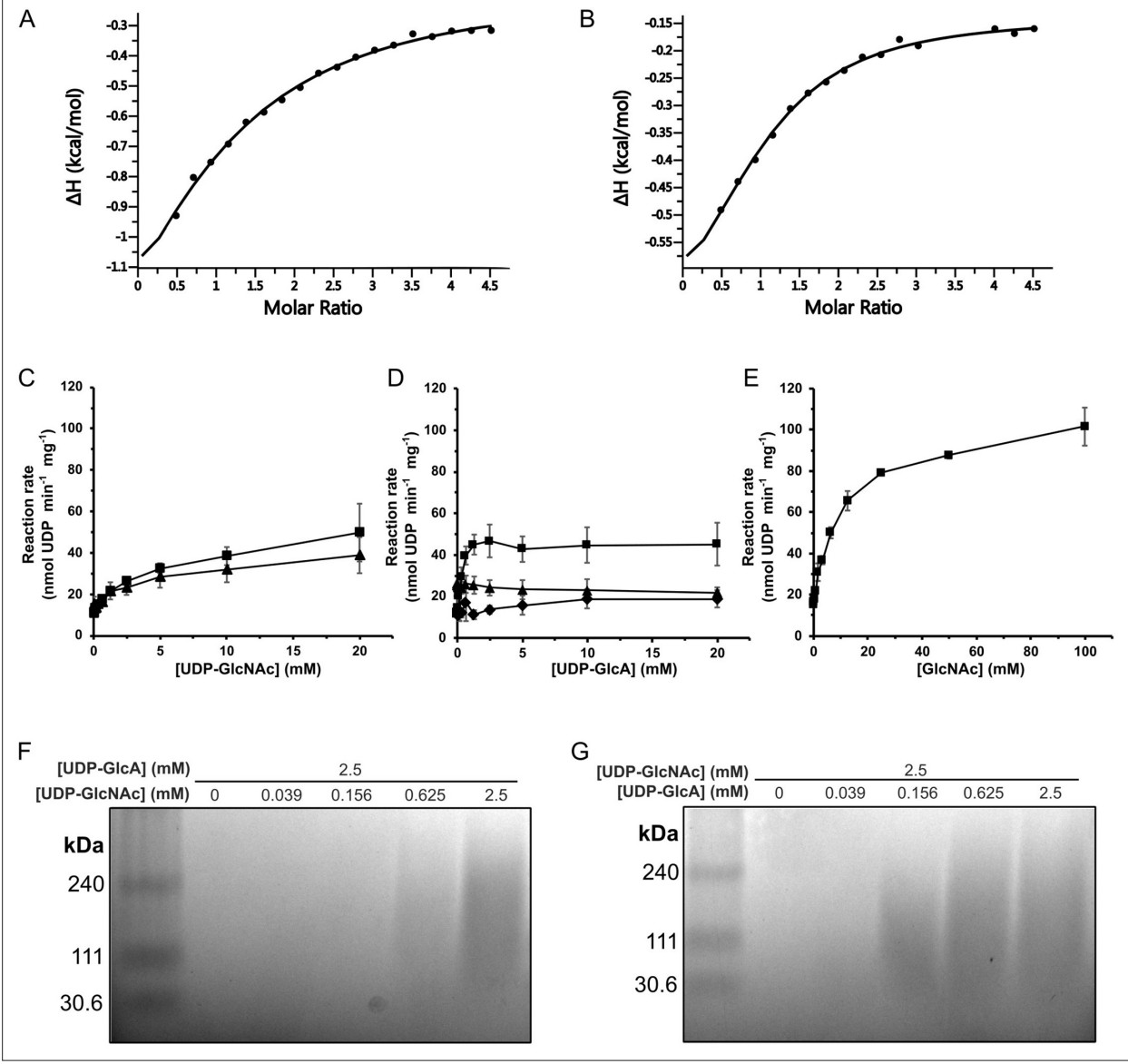

**Figure 2.** Biochemical analysis of UDP-GlcA and UDP-GlcNAc utilization by CvHAS. Binding isotherms derived from ITC experiments where CvHAS was titrated with UDP-GlcA in the absence (**A**) or presence (**B**) of GlcNAc. (**C**) Scatter plot of UDP-GlcNAc turnover without an acceptor (■) or in the presence of 2.5 mM UDP-GlcA (▲). (**D**) UDP-GlcA turnover without an acceptor (♦), in the presence of 10 mM GlcNAc (■) and in the presence of 2.5 mM UDP-GlcNAc (▲). Data points for panels **C** and **D** are reported as the average of five technical replicates. (**E**) Measurement of UDP-GlcA turnover at a constant concentration of 2.0 mM with titration of GlcNAc (■). Data points are reported as the average of three technical replicates. All error bars represent the standard deviation from the mean. (**F**) HA electrophoresis gel analyzing HA production under UDP-GlcNAc limiting conditions with excess UDP-GlcA. (**G**) HA electrophoresis gel analyzing HA production under UDP-GlcA limiting conditions with excess UDP-GlcNAc. Confirmation of HA product identity by HA lyase digestion under the provided synthesis conditions has been described in previous reports (*Górniak et al., 2025*). Molecular weight standards correspond to a Select-HA LoLadder (Echelon Biosciences).

The online version of this article includes the following source data and figure supplement(s) for figure 2:

**Source data 1.** Raw image file for Stains-All gel of CvHAS HA synthesis products under substrate limiting conditions.

**Source data 2.** Stains-All gel of CvHAS HA synthesis products under substrate limiting conditions with relevant lanes labeled.

**Figure supplement 1.** ITC plots for CvHAS substrate titration.

**Figure supplement 2.** Michaelis–Menten fits for CvHAS substrate titration.

release of UDP upon glycosyl transfer to the activities of pyruvate kinase and lactate dehydrogenase, as previously described (*Blackburn et al., 2018*; *Hubbard et al., 2012*). First, measurements of UDP-GlcA and UDP-GlcNAc turnover in the absence of an acceptor substrate were performed. We observed an incremental increase in reaction rate in response to UDP-GlcNAc titration, allowing us to derive an apparent Km of 744 µM. (*Figure 2C*, *Figure 2—figure supplement 2A*).

Second, when titrating UDP-GlcA, no apparent change in substrate turnover rate above background was observed (*Figure 2D*), as also previously described (*Maloney et al., 2022*). This suggests that CvHAS is unable to hydrolyze UDP-GlcA.

Third, a kinetic experiment reflecting the first glycosyl transfer reaction performed by CvHAS was set up. Here, excess GlcNAc was supplemented to generate a primed state, and UDP-GlcA was titrated. Under those conditions, a steep kinetic response to substrate titration was observed, with an apparent Km for UDP-GlcA of 147 µM (*Figure 2B*, *Figure 2—figure supplement 2B*).

Fourth, to compare kinetics of substrate turnover during initiation and processive HA synthesis, a separate set of experiments was performed. First, UDP-GlcNAc was titrated in the presence of excess UDP-GlcA. Here, maximum reaction velocity was approximately 25% lower than that measured in the presence of UDP-GlcNAc only; however, the substrate concentration at which saturation was reached appeared unchanged (*Figure 2C*). Second, titration of UDP-GlcA at an excess of UDP-GlcNAc revealed a constant or even slightly declining substrate turnover rate (*Figure 2D*). This finding suggests that the UDP-GlcNAc turnover rate remains the same, regardless of whether GlcA or presumably water forms the acceptor of the glycosyl transfer reaction.

To confirm that HA was synthesized under the titration conditions, we visualized the HA product by gel electrophoresis and 'Stains-All' staining. To this end, synthesis reactions were quenched after 1.5 hr and electrophoresed through an agarose gel, as previously described (*Górniak et al., 2025*). Previous work showed that HA made by CvHAS appears as a polydisperse distribution in a molecular weight range between 30 and 300 kDa. Our HA electrophoresis experiment confirmed formation of a similar product, with HA signal initially observed at a UDP-GlcNAc concentration of 625 µM (*Figure 2F*), and a UDP-GlcA concentration of 156 µM (*Figure 2G*).

## GlcNAc primer affinity

Turnover of UDP-GlcA by CvHAS requires a GlcNAc acceptor, either in the form of a primer or the non-reducing end terminal HA moiety. The enzyme-coupled assay described above allowed us to determine the apparent affinity of CvHAS for the GlcNAc primer. While only minimal UDP-GlcA hydrolysis by CvHAS is observed in the absence of a GlcNAc primer (*Maloney et al., 2022*), CvHAS shows a drastic increase in UDP-GlcA consumption in the presence of saturating GlcNAc. This is likely due to the supplemented monosaccharide serving as the acceptor. Accordingly, titrating GlcNAc into a reaction of CvHAS at a constant UDP-GlcA concentration of 2 mM revealed a saturable increase in UDP-GlcA turnover with an apparent Km of 4.0 mM (*Figure 2E*, *Figure 2—figure supplement 2C*). We note that the apparent maximum catalytic rate observed in the presence of saturating GlcNAc is about twice the rate observed in the presence of saturating UDP-GlcA and UDP-GlcNAc (*Figure 2C–E*). While the reaction in the presence of the GlcNAc primer (*Figure 2E*) likely generates disaccharides that diffuse away from the enzyme, the formation of a proper HA polymer in the presence of both UDP-activated substrates (*Figure 2C*) appears to reduce CvHAS' overall catalytic rate.

## Detergent interactions at the acceptor site

Structural and functional analyses of CvHAS are routinely performed in lipid nanodiscs or the detergent glyco-diosgenin (GDN) (*Maloney et al., 2022*). Both conditions support the catalytic activity of the enzyme. However, the protein is initially extracted from membranes in the detergent dodecyl-β-D-maltopyranoside (DDM) that is later removed during the purification procedure (see Methods). In a DDM-solubilized state, CvHAS and XlHAS-1 are catalytically inactive (*Figure 3—figure supplement 1*), yet regain activity after nanodisc reconstitution or exchange into GDN detergent.

To our surprise, careful analysis of a cryo-EM dataset in which CvHAS was supplemented with 2 mM MnCl$_2$ and 1 mM UDP-GlcA failed to converge on a substrate-bound map. However, approximately 50% of all particles that generated a high-resolution map retained a well-resolved DDM molecule near the active site (*Figure 3A*, *Figure 3—figure supplement 2*). The first glucosyl unit of the maltoside moiety stacks against Trp342 at the acceptor site. The second non-reducing glucosyl unit protrudes

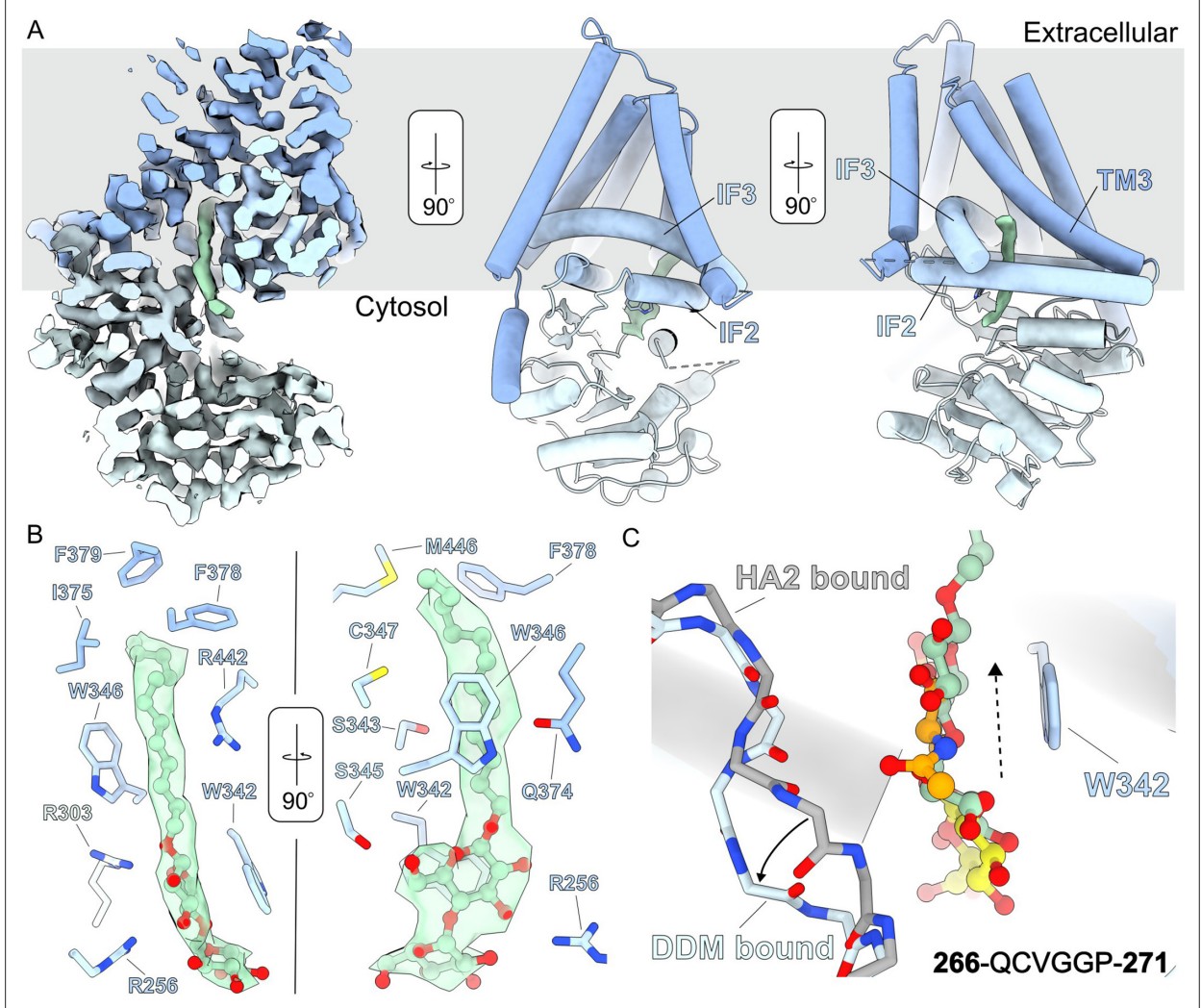

**Figure 3.** DDM binding at the acceptor site. (**A**) Left – cross-section of a cryo-EM density map for CvHAS (blue) bound to DDM (green). Middle and right – atomic coordinates of CvHAS with DDM cryo-EM density independently contoured. Relevant interfacial (IF) and transmembrane (TM) helices are labeled. (**B**) DDM coordination. DDM is shown in ball and stick representation with light green carbon atom coloring. (**C**) Structural alignment of DDM-bound and HA disaccharide (HA2)-bound (PDB ID: 8snc) coordinates. The switch loop (266-271) is displayed with backbone atoms only for clarity, with the DDM-bound conformation shown in light blue and the HA2-bound conformation shown in gray. A solid arrow is used to indicate the direction of switch loop flipping from the HA2- to DDM-bound state. A dashed arrow is used to indicate the direction of displacement of DDM's maltose group relative to the HA disaccharide.

The online version of this article includes the following figure supplement(s) for figure 3:

**Figure supplement 1.** HA synthesis activity is abolished by DDM.

**Figure supplement 2.** Cryo-EM data processing for DDM-bound CvHAS.

**Figure supplement 3.** Switch loop movement in GlcNAc, HA disaccharide, and DDM-bound CvHAS structures.

into the nucleotide-binding cleft, thereby overlapping with the donor sugar-binding site (*Figure 3B*). DDM's interaction with Trp342 is intriguing because this residue also stabilizes the GlcNAc monosaccharide in the primed state (*Maloney et al., 2022*) or the terminal GlcNAc unit in an HA-associated state (*Górniak et al., 2025*). Indeed, the maltose moiety occupies a position similar to the previously described HA disaccharide-binding pose obtained after extending a GlcNAc primer with GlcA (*Górniak et al., 2025*; *Figure 3C*).

Relative to the HA disaccharide-bound state, the reducing end glucosyl unit of DDM is shifted toward the TM channel by about 2 Å (*Figure 3C*). In this position, its C6 hydroxyl group is in hydrogen bonding distance to Ser345, and the ring oxygen interacts with the guanidinium group of Arg303,

which is about 3.1 Å away (*Figure 3B*). Additionally, a previously described 'switch' loop (residues 267–271) at the back of the substrate-binding groove (*Maloney et al., 2022*) moves approximately 3.6 Å away from the maltose moiety relative to the HA disaccharide-bound structure. This conformational change prevents direct interactions of the switch loop with DDM's head group (*Figure 3C*, *Figure 3—figure supplement 3A*).

Strikingly, DDM's dodecyl alkyl chain extends into a hydrophobic tunnel that is formed by CvHAS' transmembrane region. The tunnel is formed by Interface Helices two and three as well as TM helix 3 (*Figure 3A*). It is open to the lipid bilayer environment which likely allows lipid acyl chains to partially enter the enzyme in a biological membrane. The tunnel is lined by Ser343, Trp346, Cys347, Gln374, Ile375, Phe378, Phe379, Arg442, and Met446 (*Figure 3B*).

## Acceptor promiscuity

The intriguing binding pose of DDM's maltoside headgroup at the acceptor site prompted us to investigate whether CvHAS accepts other carbohydrates as glycosyl transfer substrates. To this end, we monitored the increase of UDP-GlcA turnover by CvHAS in the presence of selected mono- and disaccharides. As discussed, the addition of GlcNAc to a reaction of CvHAS and UDP-GlcA dramatically increases UDP-GlcA turnover, which is monitored using the enzyme-coupled reaction described above.

Under similar conditions, we tested the suitability of the monosaccharides D-galactose, D-glucose, D-mannose, L-rhamnose, and L-arabinose to prime CvHAS. None of the monosaccharides tested were able to increase UDP-GlcA turnover above the unsubstituted condition (*Figure 4—figure supplement 1A*). Next, the disaccharides cellobiose, chitobiose, maltose, sucrose, and xylobiose were tested. Of those, only cellobiose and chitobiose increased UDP-GlcA turnover at elevated concentrations between 10 and 50 mM (*Figure 4A*). Interestingly, cellotriose and cellotetraose failed to elicit a similar effect on UDP-GlcA turnover (*Figure 4—figure supplement 1A*).

In previous studies, GlcA alone failed to prime a UDP-GlcNAc transfer reaction by CvHAS, likely due to instability at the acceptor site (*Maloney et al., 2022*). Alternatively, GlcA stability may be improved when presented in the form of a non-reducing end cap of a HA oligosaccharide. To this end, we tested the effect of titrating a HA tetrasaccharide (HA4) with GlcA at the non-reducing end on UDP-GlcNAc turnover. At up to 25 mM concentration, no apparent change in UDP-GlcNAc turnover rate was observed (*Figure 4—figure supplement 1B*), perhaps due to diffusion limitations imposed by a closed HA channel. The results are consistent with previous reports on XlHAS-1 that also failed to elongate an HA tetrasaccharide (*DeAngelis, 1999*).

Compared to the GlcNAc monosaccharide, the stimulatory effects of chito- and cellobiose are reduced to about 15 and 50%, respectively, at the highest concentrations tested (50 mM, *Figure 4A*). Consistent with previous observations, no increase in UDP-GlcNAc turnover was observed in the presence of added mono- and disaccharides (*Figure 4B*). Instead, cello- and chitobiose, as well as GlcNAc, slightly reduced its turnover, compared to the unsubstituted reaction.

Increasing UDP-GlcA turnover in the presence of cellobiose or chitobiose suggests that the disaccharides may serve as acceptors of the GlcA transfer reaction. We employed thin layer chromatography (TLC) to separate the reactants and products and visualized them with thymol stain (see Methods). As shown in *Figure 4—figure supplement 2A*, UDP-GlcA, GlcA, and cellobiose are appreciably separated on Silica Gel 60 plates using a 5:3:2 volume ratio of butanol/ethanol/water as solvent. UDP-GlcA shows the least mobility and remains close to the origin position, clearly separated from GlcA. In the presence of cellobiose and UDP-GlcA, a CvHAS product species can be detected that migrates slightly above the UDP-GlcA position (*Figure 4—figure supplement 2A*).

Repeating the TLC analysis with diphenylamine staining instead of thymol provides a different colorimetric readout for cello- and chitobiose adducts (dark blue and burgundy bands). For both disaccharides, we detect putative monosaccharide adducts under reaction conditions that migrate below the cello- and chitobiose species (*Figure 4C*). The appearance of these products depends on the presence of CvHAS, UDP-GlcA, and the disaccharide, suggesting that the bands indeed represent novel reaction products of CvHAS. Conversely, replacing UDP-GlcA with UDP-GlcNAc as the substrate did not result in the formation of detectable new products with cello- or chitobiose, suggesting that these acceptors can only be extended with GlcA (*Figure 4D*).

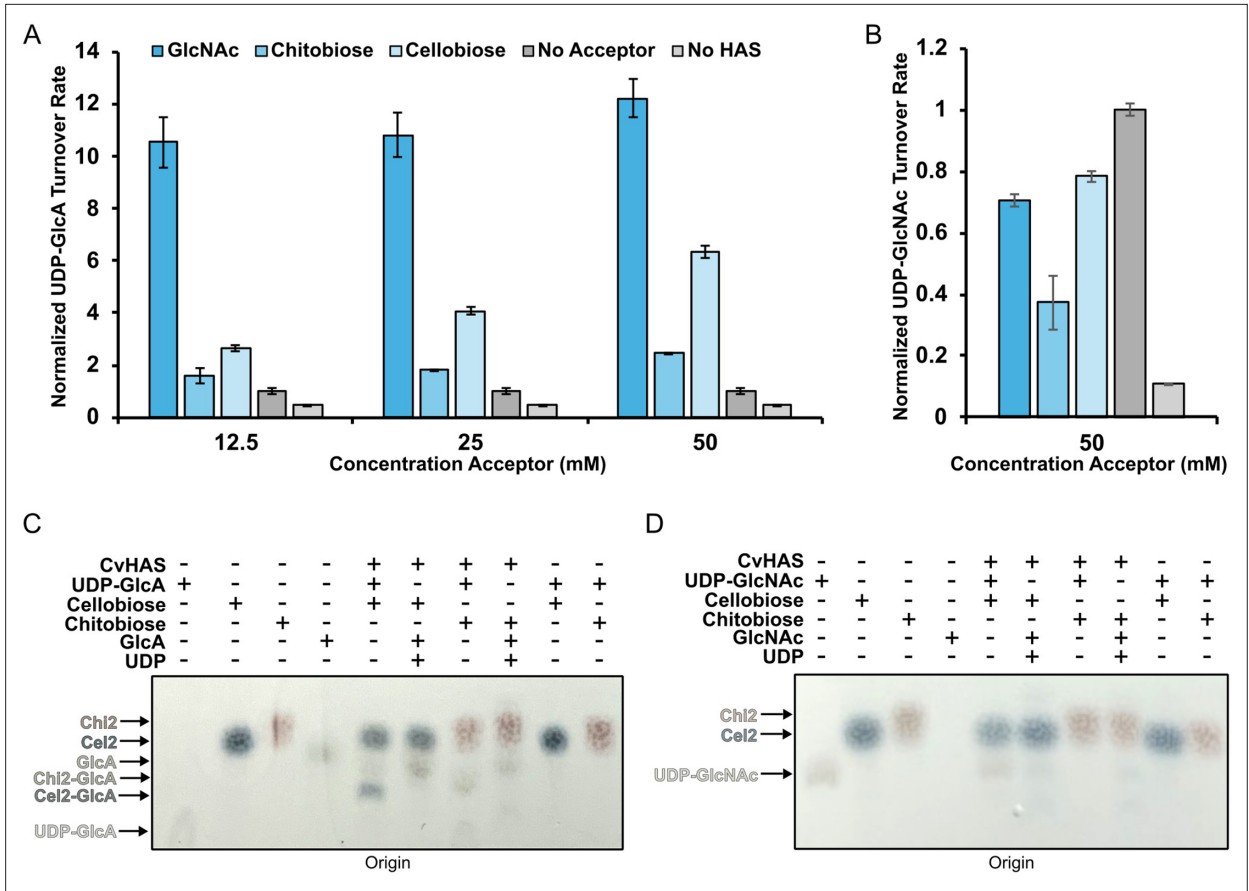

**Figure 4.** Transfer of GlcA to non-canonical acceptor substrates. Relative reaction rates for UDP-GlcA (**A**) and UDP-GlcNAc (**B**) turnover in the presence of GlcNAc, chitobiose, and cellobiose. Rates were normalized to samples where the acceptor volume was replaced with water (light gray). Individual velocities were calculated as the average of three technical replicates. Error bars correspond to the standard deviation from the mean. Diphenylamine staining of thin layer chromatography (TLC) plates developed after spotting CvHAS glycosyl transfer reaction mixtures containing either UDP-GlcA (**C**) or UDP-GlcNAc (**D**).

The online version of this article includes the following source data and figure supplement(s) for figure 4:

**Source data 1.** Raw image files for diphenylamine exposed thin layer chromatography (TLC) plate displaying cellobiose and chitobiose primer extension with GlcA.

**Source data 2.** Diphenylamine exposed thin layer chromatography (TLC) plates displaying putative cellobiose and chitobiose primer extension with GlcA.

**Figure supplement 1.** Kinetic traces of UDP-GlcA hydrolysis.

**Figure supplement 2.** Extension of cellobiose and chitobiose by GlcA.

**Figure supplement 2—source data 1.** Raw image files for visualization of cellobiose and chitobiose glycosyl transfer products with UDP-GlcA generated by CvHAS using thymol, diphenylamine, and 14C-GlcA-based labeling.

**Figure supplement 2—source data 2.** Visualization of cellobiose and chitobiose glycosyl transfer products with UDP-GlcA generated by CvHAS using thymol, diphenylamine, and 14C-GlcA-based labeling.

To confirm that the extended disaccharide species indeed contain GlcA, we repeated the synthesis reaction in the presence of 14C-labeled UDP-GlcA. TLC analysis followed by autoradiography revealed species migrating slightly above and below the GlcA monosaccharide band for chito- and cellobiose, respectively (*Figure 4—figure supplement 2C*). While the mobility of the individual bands between TLC experiments shows some variability, our data suggests that cellobiose and chitobiose can indeed serve as GlcA acceptors, albeit at high concentrations.

## Discussion

HA biosynthesis is a multi-step process. It involves the sequential binding of UDP-activated GlcA and GlcNAc to HAS's catalytic site, the transfer of the donor sugar to an acceptor, and the translocation of the nascent HA polymer across the plasma membrane (*DeAngelis and Zimmer, 2023*). Important differences exist in how CvHAS interacts with its substrate. The more abundant UDP-GlcNAc substrate is readily bound and hydrolyzed by apo CvHAS to initiate HA biosynthesis. Turnover of the second substrate, UDP-GlcA, however, is inefficient in the absence of an accepting carbohydrate moiety and does not prime the synthesis reaction.

The resolved proofreading and inserted binding poses of UDP-GlcA suggest that recruitment of this substrate occurs in multiple steps. The proofreading pose, in which the carboxylate is recognized between positively charged residues, may serve to distinguish UDP-GlcA from UDP-Glc, which is the more abundant metabolite. Upon its proper insertion into the catalytic cleft, the subtle reorganization of the surrounding priming loop likely positions and stabilizes UDP-GlcA for glycosyl transfer.

Observing two distinct UDP-sugar binding poses is not entirely unprecedented for processive GTs. A comparison of cryo-EM structures for *P. sojae* and *C. albicans* chitin synthase (CHS) in complex with UDP-GlcNAc revealed that CHSs likely utilize an analogous mechanism for substrate insertion (*Chen et al., 2022*; *Ren et al., 2022*). In one case, the GlcNAc moiety points almost 180° away from the active site and must therefore undergo large spatial rearrangements to adopt its proper position for glycosyl transfer (*Figure 1—figure supplement 4*).

It is currently unclear why GlcA cannot prime HA biosynthesis. The turnover of UDP-GlcA is substantially increased in the presence of an accepting glycosyl unit, likely due to the positioning of a suitable nucleophile to mediate the attack on the donor sugar. Because the apparent dissociation constants for UDP-GlcA in the absence and the presence of GlcNAc are similar, a GlcNAc primer has only modest effects on substrate binding but significantly affects the stability of the substrate at the active site.

Due to exposure during purification, CvHAS accommodates a DDM molecule at its acceptor site and an adjacent hydrophobic tunnel. DDM's dodecyl tail, which likely mimics a phospholipid acyl chain, seals off a lateral connection to the active site. Coordination of DDM's reducing end glucosyl unit prompted us to investigate CvHAS' promiscuity toward alternative primers. Indeed, our findings revealed CvHAS can catalyze unusual biochemistry through the addition of GlcA to cellobiose or chitobiose, albeit at high concentrations. This finding points to degeneracy in the active site of HAS, relaxing specificity toward acceptor substrates. Evolutionary conservation with other QXXRW motif-bearing GTs, namely cellulose and chitin synthases, could explain this phenomenon (*Bi et al., 2015*; *Stephens et al., 2023*). Both the DDM-bound structure and observations of non-canonical primer extension point to the exploitable potential for HAS and other related GTs to catalyze unnatural carbohydrate synthesis reactions.

Taken together, this work elucidates the molecular basis for dual-substrate selectivity by a model class 1-NR HAS. Visualization of UDP-GlcA in the proofreading conformation establishes a previously uncharacterized checkpoint that supports integrity of HA synthesis. In addition, this study bolsters the previously asserted hypothesis that CvHAS initiates HA synthesis via self-catalyzed formation of a GlcNAc primer through distinguishing acceptor-dependent turnover kinetics for UDP-GlcA from UDP-GlcNAc. Findings reported here should inform future investigations of other similar bifunctional GTs, such as the cellulose synthase-like (Csl) enzyme class involved in hemicellulose biosynthesis (*Pauly et al., 2013*).

## Methods

### CvHAS expression and membrane harvest

A glycerol stock of *E. coli C43(DE3)* cells harboring a pET28a-CvHAS expression plasmid (*Blackburn et al., 2018*; *Maloney et al., 2022*) was used to inoculate LB broth supplemented with 50 µg/ml kanamycin and grown overnight. The next day, 10 ml of overnight culture was added to 1 l of TB supplemented with 50 µg/ml kanamycin, 4% glycerol, and 1X M salts. Expression cultures were grown to $OD_{600} = 0.8$ at 30°C with 220 RPM shaking and cooled to 20°C before induction with 0.5 mM IPTG. Protein expression was allowed to occur overnight before harvesting cell pellets by centrifugation at $4000 \times g$ for 10 min.

Hereafter, all steps were performed at 4°C unless stated otherwise. Cell pellet taken from 4 l of expression culture was resuspended in RB containing 20 mM Tris-HCl pH 7.5, 100 mM NaCl, 10% glycerol, 0.5 mM tris(2-carboxyethyl)phosphine (TCEP). Lysozyme was added to 1 mg ml$^{-1}$ final concentration and the suspension was mixed for 30 min. The cell suspension was disrupted by three rounds of microfluidization at 18,000 PSI, with 1 mM phenylmethylsulfonyl fluoride (PMSF) added after the first passage. Crude lysate was spun at 20,000 × $g$ for 10 min to remove intact cells and larger debris. Supernatant from the first spin was subjected to a second round of centrifugation at 200,000 × $g$ for 2 hr. The resulting membrane pellet was harvested and flash-frozen in liquid nitrogen prior to CvHAS purification.

## CvHAS purification and reconstitution

CvHAS purification followed a previously described protocol (*Maloney et al., 2022*). Membrane pellet was resuspended in 120 ml of SB containing 20 mM Tris-HCl pH 7.5, 300 mM NaCl, 10% glycerol, 1% DDM, 0.1% cholesteryl hemisuccinate (CHS), and 0.5 mM TCEP. Subsequently, 1 mM PMSF was added to the membrane suspension and mixed for 1 hr. Following centrifugation at 200,000 × $g$ for 30 min, the supernatant was harvested and batch bound to 5 ml of Ni-NTA resin.

The resin–protein mixture was loaded onto a Kimble flex column and collected by gravity. When purifying CvHAS for functional assays, resin was washed with 20 CVs of WB1 containing 20 mM Tris-HCl pH 7.5, 1 M NaCl, 40 mM imidazole, 10% glycerol, 0.05% glycodiosgenin (GDN), 0.5 mM TCEP and 20 CVs of WB2 containing 20 mM Tris-HCl pH 7.5, 300 mM NaCl, 80 mM imidazole, 10% glycerol, 0.02% GDN, 0.5 mM TCEP. CvHAS was eluted in 5 CVs EB (WB2 + 300 mM imidazole). Elutions were concentrated using a 50 kDa MWCO Amicon Ultra Centrifugal Spin Filter (Millipore-Sigma) and injected onto an S200 Increase 10/300 GL (Cytiva) size exclusion column equilibrated in GFB1 containing 20 mM Tris-HCl pH 7.5, 100 mM NaCl, 0.02% GDN, 0.5 mM TCEP.

Nanodisc reconstitution was also performed as previously described (*Maloney et al., 2022*). In short, the GDN composition of purification buffers was modified to 0.02% DDM/0.002% CHS. Following size exclusion, fractions containing CvHAS were concentrated and batch mixed 1:3:3:3:30 with Nb872:Nb881:MSP1E3D1:*E. coli* total lipid extract solubilized in DDM. After 30 min, ~50 mg of hydrated SM2 adsorbent biobeads (Bio-Rad) were added to the mixture. This was repeated once after an additional 30-min incubation and again after overnight incubation. The reconstitution mixture was cleared of biobeads by filtration and re-injected on an S200 Increase 10/300 GL column equilibrated in GFB2 containing 20 mM Tris-HCl pH 7.5, 100 mM NaCl, and 0.5 mM TCEP. Fractions having CvHAS nanodiscs in complex with Nb872 and Nb881 were identified via Coomassie staining and used in cryo-EM sample preparation.

## Nanobody expression and purification

*E. coli WK506* harboring a pMES4-Nb expression plasmid (*Maloney et al., 2022*) were inoculated from glycerol stocks into LB broth supplemented with 100 µg/ml ampicillin and 1 mM MgCl$_2$ and grown overnight. Two ml of overnight culture was used to inoculate TB supplemented with 100 µg/ml ampicillin, 1X M salts, 1 mM MgCl$_2$, and 0.1% D-glucose. Cells were grown to OD$_{600}$ = 0.8 at 37°C with 220 RPM shaking, and IPTG was added to a 1 mM final concentration. The shaker temperature was dropped to 27°C, and protein expression was allowed to occur overnight. Cell pellets were harvested the next day by centrifugation at 4000 × $g$ for 10 min.

Nanobodies were periplasmically extracted by mixing the cell pellet for 30 min with hyperosmotic TES buffer containing 20 mM Tris-HCl pH 8.0, 500 mM sucrose, and 0.05 mM EDTA. The extraction mixture was diluted threefold in 0.25X TES buffer and mixed for an additional 30 min before centrifugation at 200,000 × $g$ for 30 min. The resulting supernatant was batch bound for 1 hr with 5 ml Ni-NTA resin pre-equilibrated in TBS.

Nickel-NTA resin was collected by gravity, washed with 20 CVs Nb-WB1 containing 20 mM Tris-HCl pH 8.0, 1 M NaCl, 20 mM imidazole and with 20 CVs Nb-WB2 containing 20 mM Tris-HCl pH 8.0, 100 mM NaCl, 40 mM imidazole. Nanobodies were eluted in WB2 supplemented with 300 mM imidazole. Nickel elutions were concentrated over a 10 kDa MWCO centrifugal spin filter prior to injection on an S75 HiLoad size exclusion column equilibrated in Nb-GFB containing 20 mM Tris-HCl pH 7.5, 100 mM NaCl. SEC fractions with nanobodies were pooled and flash frozen for CvHAS reconstitution.

## Cryo-EM sample preparation

For capturing CvHAS in complex with UDP-GlcA, a catalytically inactive mutant of CvHAS with an Asn substitution for Asp302 was used, as described previously (*Blackburn et al., 2018*). The inactive CvHAS–Nb complex was concentrated to 4.0 mg/ml and supplemented with 10 mM $MnCl_2$ and 5 mM UDP-GlcA. The mixture was incubated for 15 min on ice, after which 3.0 µl was applied to glow-discharged QF R1.2/1.3 300 mesh Cu grids and blotted for 6 s at 4°C/100% humidity. Grids were plunged into liquid ethane using a Mark IV Vitrobot (Thermo Fisher). For the DDM-bound map, D302N CvHAS was incubated with 2 mM $MnCl_2$ and 1 mM UDP-GlcA for 15 min prior to freezing.

## Cryo-EM data collection and processing

For UDP-GlcA-bound structures, all imaging was done on a Titan Krios equipped with a K3 direct electron detector and GIF energy filter at UVA's Molecular Electron Microscopy Core. Imaging was performed in counting mode at a calibrated pixel resolution of 1.08 Å using a 10 eV slit width, with a target defocus range of –2.0 to –1.0 µm and total dose of 50 $e^-/Å^2$.

Movies were imported to cryoSPARC v4.0.3 (*Punjani et al., 2017*) for Patch Motion Correction and Patch CTF Estimation. Particles were initially selected by blob picking and extracted as inputs for 2D template generation. Particles from template picking were extracted with a box size of 256 pixels and 2X Fourier cropping. A subset of the initial picks was used to generate three volume references through ab initio reconstruction, converging on one reliable CvHAS–Nb complex volume and two noise volumes. Iterative heterogeneous refinement with the ab initio references was used to remove bad picks. The curated particle set was re-extracted at full box size and used for non-uniform refinement of the initial good volume. The refined particles were passed to a 3D classification job with a focus mask covering the GT domain. Classes containing density for UDP-GlcA were carried over for a second round of 3D classification using the same masking approach. Two classes corresponding to inserted and proofreading states were identified and independently processed using non-uniform and local refinement jobs.

For the DDM-bound structure, imaging of a UDP-GlcA containing sample was performed at a calibrated pixel resolution of 0.652 Å. A target defocus range of –0.8 to –1.8 µm and total dose of 60 $e^-/Å^2$ were used. Initial particle picking and curation workflow followed that described for the UDP-GlcA dataset; however, particles were extracted at a 400 pixel box size and binned 4X for all steps preceding the first round of non-uniform refinement. Particles were re-extracted at full box size prior to running non-uniform refinement, global/local CTF refinement, and local refinement to arrive at a final reconstruction with well-resolved DDM density.

## Model building and refinement

An initial model for both inserted UDP-GlcA and UDP-GlcA proofreading structures was taken from PDB ID: 8snd. GlcNAc was deleted manually in COOT (*Emsley and Cowtan, 2004*). For building coordinates with DDM bound, PDB ID: 7sp9 was selected as the initial model. GlcNAc was deleted from the initial model and DDM (Monomer ID: LMT) was imported and fit into the cryo-EM density map in COOT. All three models were iteratively real-space refined in COOT and Phenix (*Adams et al., 2010*; *Zwart et al., 2008*).

## Isothermal titration calorimetry

CvHAS was purified with 10 mM MnCl2 supplemented in GFB1. Three hundred µl of 45 µM CvHAS was loaded into the sample cell of a MicroCal PEAQ-ITC (Malvern Panalytical). For injection series without GlcNAc, a syringe concentration of 1 mM UDP-GlcA dissolved in the modified GFB1 was used. The same approach was followed for measuring UDP, UDP-Glc, and UDP-GlcNAc binding. For injection series with GlcNAc, 10 mM GlcNAc and 10 mM $MnCl_2$ were included in GFB1 for CvHAS purification and UDP-GlcA dissolution. A 20-injection series with 2.0 µl injection volumes and a reference power of 5 µcal/s was carried out. Data fitting and binding constant (Kd) derivation was performed using MicroCal PEAQ-ITC analysis software.

## Enzyme-coupled substrate turnover assays

Enzyme-coupled UDP quantification was performed as previously described (*Blackburn et al., 2018*; *Omadjela et al., 2013*). A stock reaction mix containing 40 mM Tris-HCl pH 7.5, 150 mM NaCl, 20 mM

$MnCl_2$, 1 mM TCEP, 1.5 mM NADH, 2 mM phosphoenol pyruvate, and 2 U/µl Lactate Dehydrogenase/Pyruvate Kinase enzyme cocktail (Sigma) was prepared. A twofold dilution series beginning at 20 mM for both UDP-GlcNAc and UDP-GlcA was performed across 12 wells of a 96-well black, microclear bottom plate (Grenier). The stock reaction mix was diluted twofold with UDP-sugar well solution and 1 µM CvHAS to initiate the reaction. Absorbance was measured at $\lambda$ = 340 nm each minute over a 1.5- to 3-hr period at 30°C. Initial reaction velocities were taken as the slope of a linear regression fit to data points between 30 and 60 min of each kinetic trace. Non-linear Michaelis–Menten fitting was performed with Prism 6.0. When measuring the influence of alternative glycosyl transfer acceptors on substrate turnover, UDP-GlcA or UDP-GlcNAc was included in the stock reaction mix at 4.0 mM. Concentrations of monosaccharides and disaccharides used for screening were varied between 12.5 and 50 mM.

## HA gel electrophoresis

A 1% ultrapure agarose gel (Sigma) was cast with 1X TAE buffer. Reactions were set up as described for substrate hydrolysis measurements and quenched after 1.5 hr with Laemmli buffer. Agarose gel electrophoresis was run at 100 V for 2 hr in 1X TAE. The resulting gel was fixed in 50% EtOH for 1 hr and subsequently placed in 0.005% Stains-All dissolved in 50% EtOH overnight under dark. The stained gel was transferred to a 20% EtOH solution and again left overnight. The next day, the gel was exposed to ambient light to remove remaining Stains-All background prior to imaging, as described (*Górniak et al., 2025*).

## HA quantitation by paper chromatography and liquid scintillation counting

Radiometric HA quantification was performed as described (*Hubbard et al., 2012*). HA synthesis reactions were initiated by mixing 10 µM CvHAS 1:1 with a reaction mix containing 80 mM Tris-HCl pH 7.5, 150 mM NaCl, 1 mM TCEP, 40 mM $MnCl_2$, 10 mM UDP-GlcA, 10 mM UDP-GlcNAc, 0.1 µCi 3 H-UDP-GlcNAc (Revvity). Reactions were allowed to occur for 2 hr at 30°C before quenching with 2% SDS. Samples were spotted on Whatman filter paper and dried. The paper was developed in a 65% 1 M ammonium acetate/35% EtOH mobile phase for 2 hr and dried. The origin was extracted for liquid scintillation counting in a Hidex 300 SL by exposure to UltimaGold scintillation fluid (Revvity).

## TLC of CvHAS reaction products

Reactions for TLC were carried out by mixing 10 µM CvHAS 1:1 with 80 mM Tris-HCl pH 7.5, 150 mM NaCl, 20 mM $MnCl_2$, 1.0 mM TCEP and 10 mM UDP-GlcA, followed by incubation at 30°C for 2 hr. To observe acceptor extension, cellobiose and chitobiose were included in the reaction mix at 10–30 mM final concentration. Each reaction was mixed 1:1 with 50% MeOH, and 2.0 µl was spotted on a Silica Gel 60 plate.

TLCs were developed in a *n*-butanol/ethanol/water (5:3:2, vol/vol/vol) solvent system, dried and stained by exposure to either 0.5% thymol (wt/vol) dissolved in 50:1 $EtOH/H_2SO_4$ (*Schulze et al., 2017*; *Umekawa et al., 2022*) or 16.7% diphenylamine (wt/vol) dissolved in 4:3:17 aniline/$H_3PO_4$/acetone (*Zhang et al., 2007*). For analysis of radioactive products, 0.02 µCi of 14 C-UDP-GlcA was included in the reaction mix. Autoradiography was performed by exposing the TLC plate to a phosphor screen for 2 days prior to phosphor imaging on a Typhoon IP instrument (Amersham).

## Site-directed mutagenesis

Complementary forward and reverse primers with the integrated mutant codon annealing at the mutation site, as well as forward and reverse primers for the T7 promoter and T7 terminator sequence of the pET28a-CvHAS vector were generated (IDT). For each mutation, three PCRs amplifying the region of the plasmid sequence from T7 promoter to mutation site, mutation site to T7 terminator, and T7 terminator to T7 promoter were carried out in parallel. Resulting amplicons were purified by gel extraction (QIAGEN) and used in a HIFI reaction (NEB) for plasmid assembly. The assembly reactions were transformed into chemically competent DH5α cells, followed by DNA purification and full-plasmid sequence verification.

## Acknowledgements

We are grateful to Michael Purdy and David Cooper at the Molecular Electron Microscopy Core facility for help and support in cryo-EM data collection. Additional thanks are extended to Ireneusz Gorniak for providing purified XlHAS-1. The project was in part funded by NIH grant R35GM144130. JZ is an Investigator of the Howard Hughes Medical Institute. This article is subject to HHMI's Immediate Access to Research policy, which requires that this article be made publicly available as initial and revised preprints deposited on a designated preprint server under a CC BY 4.0 license.

## Additional information

### Funding

| Funder | Grant reference number | Author |
|---|---|---|
| National Institutes of Health | | Zachery Stephens |
| Howard Hughes Medical Institute | | Jochen Zimmer |
| National Institutes of Health | R35GM144130 | Zachery Stephens |

The funders had no role in study design, data collection, and interpretation, or the decision to submit the work for publication.

### Author contributions

Zachery Stephens, Conceptualization, Investigation, Writing – review and editing; Julia Karasinska, Investigation, Writing – review and editing; Jochen Zimmer, Conceptualization, Resources, Formal analysis, Writing – original draft

### Author ORCIDs

Zachery Stephens https://orcid.org/0009-0001-0140-5304
Julia Karasinska https://orcid.org/0009-0008-4298-9159
Jochen Zimmer https://orcid.org/0000-0002-8423-2882

Reviewer #1 (Public review): https://doi.org/10.7554/eLife.109624.3.sa1
Reviewer #2 (Public review): https://doi.org/10.7554/eLife.109624.3.sa2
Author response https://doi.org/10.7554/eLife.109624.3.sa3

## Additional files

### Supplementary files

MDAR checklist

Supplementary file 1. Cryo-EM data collection, refinement, and validation statistics.

Supplementary file 2. DNA oligonucleotide primers used for CvHAS mutagenesis.

### Data availability

Cryo-EM maps have been deposited in the EMDB under the accession codes EMD-73321, EMD-73323, and EMD-73324. Protein coordinates have been deposited in the PDB under the accession codes 9YQ2, 9YQ4, and 9YQ5.

The following datasets were generated:

| Author(s) | Year | Dataset title | Dataset URL | Database and Identifier |
|---|---|---|---|---|
| Stephens Z, Zimmer J | 2025 | Chlorella virus hyaluronan synthase bound to DDMCryoEM | https://www.ebi.ac.uk/emdb/EMD-73321 | EMDataBank, EMD-73321 |
| Stephens Z, Zimmer J | 2025 | Chlorella virus hyaluronan synthase bound to a proofreading UDP-GlcA | https://www.ebi.ac.uk/emdb/EMD-73323 | EMDataBank, EMD-73323 |
| Stephens Z, Zimmer J | 2025 | Chlorella virus hyaluronan synthase bound to an inserted UDP-GlcA | https://www.ebi.ac.uk/emdb/EMD-73324 | EMDataBank, EMD-73324 |
| Stephens Z, Zimmer J | 2025 | Chlorella virus hyaluronan synthase bound to an inserted UDP-GlcA | https://doi.org/10.2210/pdb9yq5/pdb | Worldwide Protein Data Bank, 10.2210/pdb9yq5/pdb |
| Stephens Z, Zimmer J | 2025 | Chlorella virus hyaluronan synthase bound to DDM | https://doi.org/10.2210/pdb9yq2/pdb | Worldwide Protein Data Bank, 10.2210/pdb9yq2/pdb |
| Stephens Z, Zimmer J | 2025 | Chlorella virus hyaluronan synthase bound to a proofreading UDP-GlcA | https://doi.org/10.2210/pdb9yq4/pdb | Worldwide Protein Data Bank, 10.2210/pdb9yq4/pdb |

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
