## [Editor Report · eLife Assessment]

This study addresses a **fundamental** question in glycobiology by elucidating how a single-site processive enzyme orchestrates the alternating addition of sugars to synthesize complex polysaccharides such as hyaluronan. The findings are **compelling**, providing a clear mechanistic framework supported by strong experimental validation. Major strengths include the integration of high-resolution structural data with rigorous biochemical analyses, resulting in a well-supported model of hyaluronan assembly.

---

## [Referee Report · Reviewer #1 (Public review)]

Summary:

This revised manuscript describes critical intermediate reaction steps of a HA synthase at the molecular level; specifically, they examine the 2nd step, polymerization, adding GlcA to GlcNAc to form the initial disaccharide of the repeating HA structure. Unlike the vast majority of known glycosyltransferases, the viral HAS (a convenient proxy extrapolated to resemble the vertebrate forms) uses a single pocket to catalyze both monosaccharide transfer steps. The authors work illustrates the interactions needed to bind & proof-read the UDP-GlcA using direct and '2nd layer' amino acid residues. This step also allows the HAS to distinguish the two UDP-sugars; this is very important as the enzymes are not known or observed to make homopolymers of only GlcA or GlcNAc, but only make the HA disaccharide repeats GlcNAc-GlcA.

Strengths:

Techniques & analysis; overview of HA synthase mechanisms

Weaknesses:

None

Comments on revisions:

Previous clarity issues in the original submission were all resolved. Again, this is a very well done body of work!!

---

## [Referee Report · Reviewer #2 (Public review)]

Summary:

The paper by Stephens and co-workers provides important mechanistic insight into how hyaluronan synthase (HAS) coordinates alternating GlcNAc and GlcA incorporation using a single Type-I catalytic centre. Through cryo-EM structures capturing both "proofreading" and fully "inserted" binding poses of UDP-GlcA, combined with detailed biochemical analysis, the authors show how the enzyme selectively recognizes the GlcA carboxylate, stabilizes substrates through conformational gating, and requires a priming GlcNAc for productive turnover.

These findings clarify how one active site can manage two chemically distinct donor sugars while simultaneously coupling catalysis to polymer translocation.

The work also reports a DDM-bound, detergent-inhibited conformation that possibly illuminates features of the acceptor pocket, although this appears to be a purification artefact (it is indeed inhibitory) rather than a relevant biological state.

Overall, the study convincingly establishes a unified catalytic mechanism for Type-I HAS enzymes and represents a significant advance in understanding HA biosynthesis at the molecular level.

Strengths:

There are many strengths.

This is a multi-disciplinary study with very high-quality cryo-EM and enzyme kinetics (backed up with orthogonal methods of product analysis) to justify the conclusions discussed above.

Comments on revisions:

The suggestions made in the initial comments have all been responded to very well.

---

## [Author Response]

The following is the authors’ response to the original reviews.

**Public Reviews:**

**Reviewer #1 (Public review):**
Summary:This manuscript describes critical intermediate reaction steps of a HA synthase at the molecular level; specifically, it examines the 2nd step, polymerization, adding GlcA to GlcNAc to form the initial disaccharide of the repeating HA structure. Unlike the vast majority of known glycosyltransferases, the viral HAS (a convenient proxy extrapolated to resemble the vertebrate forms) uses a single pocket to catalyze both monosaccharide transfer steps. The authors' work illustrates the interactions needed to bind & proof-read the UDP-GlcA using direct and '2nd layer' amino acid residues. This step also allows the HAS to distinguish the two UDP-sugars; this is very important as the enzymes are not known or observed to make homopolymers of only GlcA or GlcNAc, but only make the HA disaccharide repeats GlcNAc-GlcA.Strengths:Overall, the strengths of this paper lie in its techniques & analysis.The authors make significant leaps forward towards understanding this process using a variety of tools and comparisons of wild-type & mutant enzymes. The work is well presented overall with respect to the text and illustrations (especially the 3D representations), and the robustness of the analyses & statistics is also noteworthy.Furthermore, the authors make some strides towards creating novel sugar polymers using alternative primers & work with detergent binding to the HAS. The authors tested a wide variety of monosaccharides and several disaccharides for primer activity and observed that GlcA could be added to cellobiose and chitobiose, which are moderately close structural analogs to HA disaccharides. Did the authors also test the readily available HA tetramer (HA4, [GlcA-GlcNAc]2) as a primer in their system? This is a highly recommended experiment; if it works, then this molecule may also be useful for cryo-EM studies of CvHAS as well.

The reviewer requested testing whether an HA tetratsaccharide could also serve as an glycosyl transfer acceptor for HAS. The commerically available HA tetrasaccharide (HA4) is terminated at its non-reducing end by GlcA, therein we proceeded to measure its effect on UDP-GlcNAc turnover kientics. Titration of HA4 failed to elicit any detectable change in UDP-GlcNAc turnover rate, indicating no priming. This is now mentioned in the main text and the data is shown in Fig. S9.

Weaknesses:In the past, another report describing the failed attempt of elongating short primers (HA4 & chitin oligosaccharides larger than the cello- or chitobiose that have activity in this report) with a vertebrate HAS, XlHAS1, an enzyme that seems to behave like the CvHAS (https://pubmed.ncbi.nlm.nih.gov/10473619/); this work should probably be cited and briefly discussed. It may be that the longer primers in the 1999 paper and/or the different construct or isolation specifics (detergent extract vs crude) were not conducive to the extension reaction, as the authors extracted recombinant enzyme.

We apologize for the oversight. This reference is now cited (ref. 18) together with the description of the failed elongation of HA4 by CvHAS.

There are a few areas that should be addressed for clarity and correctness, especially defining the class of HAS studied here (Class I-NR) as the results may (Class I-R) or may not (Class II) align (see comment (a) below), but overall, a very nicely done body of work that will significantly enhance understanding in the field.

Done as requested

**Reviewer #2 (Public review):**
Summary:The paper by Stephens and co-workers provides important mechanistic insight into how hyaluronan synthase (HAS) coordinates alternating GlcNAc and GlcA incorporation using a single Type-I catalytic centre. Through cryo-EM structures capturing both "proofreading" and fully "inserted" binding poses of UDP-GlcA, combined with detailed biochemical analysis, the authors show how the enzyme selectively recognizes the GlcA carboxylate, stabilizes substrates through conformational gating, and requires a priming GlcNAc for productive turnover.These findings clarify how one active site can manage two chemically distinct donor sugars while simultaneously coupling catalysis to polymer translocation.The work also reports a DDM-bound, detergent-inhibited conformation that possibly illuminates features of the acceptor pocket, although this appears to be a purification artefact (it is indeed inhibitory) rather than a relevant biological state.Overall, the study convincingly establishes a unified catalytic mechanism for Type-I HAS enzymes and represents a significant advance in understanding HA biosynthesis at the molecular level.Strengths:There are many strengths.This is a multi-disciplinary study with very high-quality cryo-EM and enzyme kinetics (backed up with orthogonal methods of product analysis) to justify the conclusions discussed above.Weaknesses:There are few weaknesses.The abstract and introduction assume a lot of detailed prior knowledge about hyaluronan synthases, and in doing so, risk lessening the readership pool.A lot of discussion focuses on detergents (whose presence is totally inhibitory) and transfer to non-biological acceptors (at high concentrations). This risks weakening the manuscript.

The abstract and parts of the introduction have been revised to address the reviewer’s concerns.

**Reviewer #1 (Recommendations for the authors):**
(1) As noted above, please state in title, abstract & introduction that this work is focused on a "Class I-NR HAS" (as described in Ref. #4), and NOT all HAS families...this is truly essential to note as someone working with the Pasteurella HAS version (Class II) would be totally misled & at this point, no one knows the Streptococcus HAS (Class-IR) mechanistic details which could be different due to its inverse molecular directionality of elongation compared to the CvHAS Class I-NR enzyme.

Done as requested.

(2) Page 6 - for the usefulness of the HAS mutants as being folded correctly, it was stated these mutants are suitable since they all 'purify' similarly...the use of the more proper term should probably be 'chromatograph', similarly suggesting similar hydrodynamic radii without massive folding issues.

This has been revised to state that they all exhibited comparable size exclusion chromatography profiles.

“All mutants share similar size exclusion chromatography profiles with the WT enzyme, suggesting that the substitutions do not cause a folding defect (Fig. S3).”

(3) Page 7 - please check these sentences (& rest of paragraph?) as the meaning is not clear. "First, UDP-GlcNAc was titrated in the presence of excess UDP-GlcA, resulting in a response similar to the acceptor-free condition (Fig. 2C). However, the maximum reaction velocity at 20 mM UDP-GlcNAc was approximately 25% lower than that measured in the presence of UDP-GlcNAc only (Fig. 2C)."

The paragraph has been revised to avoid confusion.

(4) In Methods, please use an italicized 'g' for the centrifugation steps globally.

Changed as requested

(5) Please note the source/vendor for the HA standards on gels.

Done

(6) Page 35 - TLC section.(a) 'n-butanol' (with italic n) is the most widespread chemical name (not butan-1-ol).

Done

(b) Also, for all of the TLC images, the origin and the solvent front should be marked.

Changed as suggested.

**Reviewer #2 (Recommendations for the authors):**
A number of minor issues should be addressed.(1) AbstractTwo comments on the Abstract, which I found surprisingly weak given the quality of the work, and lacking a key detail.A major conceptual contribution of this work is the demonstration of how a single Type-I catalytic centre discriminates, positions, and transfers two chemically distinct substrates in an alternating pattern. This distinguishes HAS from dual-active-site (Type-II) glycosyltransferases and is important for understanding HA polymerization.However, this central point is not clearly articulated in the abstract. I suggest explicitly stating that HAS performs both GlcNAc and GlcA transfer reactions within a single catalytic site, and that the proofreading/inserted poses illuminate how this multifunctionality is achieved.The abstract currently ends with the observation of a DDM-bound, detergent-inhibited state. While this is interesting, it absolutely does not represent the central conceptual advance of the study and gives the abstract an artefactual ending.I strongly recommend revising the final sentences to emphasize the broader mechanistic insight and not an "artefact" (indeed, the enzyme is inactive in the presence of this detergent; it is thus a very unusual way to conclude an abstract).That is, finish with the wider implications of how HAS coordinates alternating substrate use, proofreading, and polymer translocation. Ending on the main mechanistic or biological significance would make the abstract considerably stronger and more aligned with the main message of the paper.

The abstract has been revised thoroughly to reflect the important insights gained on CvHAS’ catalytic function and HA biogenesis in general.

(2) IntroductionThe distinction between single active-centre enzymes, which transfer both sugars alternately, and twin catalytic domain enzymes that each perform one addition is surely central to the whole paper. But it is not discussed. Surely this has to be covered. There is a lot of work in this space, including, but not limited to:
https://doi.org/10.1093/glycob/cwg085

https://doi.org/10.1093/glycob/10.9.883
https://doi.org/10.1093/glycob/cwad075 (includes this author team)Originally back to https://doi.org/10.1021/bi990270yIf the authors instead assume such a level of knowledge for the reader, then surely they are writing for a specialist audience, not consistent with the wider readership ambitions of eLife?

The Introduction has been revised as suggested by the reviewer, providing necessary background to frame our description of the Chlorella virus HAS. We made a deliberate effort to put new insights into a broader context.

(3) Results and DiscussionDDM "was observed for >50% of the analysed particles". I struggled with this. I couldn't understand how the authors selected particles that did or did not contain DDM. The main body text states: "To our surprise, careful sorting of the UDP-GlcA supplemented cryo EM dataset revealed a CvHAS subpopulation that was not bound to the substrate, but, instead, a DDM molecule near the active site (Fig 3A and S7). This was observed for >50% of the analyzed particles."That reads like there is one sample with two populations. But the figures and the methods section suggest differently: they suggest two samples with different data-collection regimes. That does not match the main text. Could this be clarified?

Yes, that wasn’t explained well. We clarified the text to stress that the DDM-bound sample came from a dataset that was intended to resolve an UDP-GlcA-bound state, but instead revealed the inhibition by DDM.

Also in this space, in the modern world, "nominal magnification" has no real meaning, and calibrated pixel size would be more appropriate. Can this be given, please?

The relevant Methods section now states: “imaging of … was performed at a calibrated pixel size of 0.652 Å”.

The discovery of DDM in the active site is surprising. But it is an inhibitory artefact. Is this section pushed a little too hard? Also, "The coordination of DDM's maltoside moiety, an αlinked glucose disaccharide, is consistent with priming by cellobiose and chitobiose." I'm not sure why an α-linked maltose is consistent with the binding of a β-linked cellobiose. That makes no sense. There will be no other enzymes where starch and cellulose oligos are mutually accepted. Consider rewriting.

We like to stress the DDM coordination because it could lead to the development of compounds that can really function as inhibitors, either for HAS or other related enzymes. In the observed DDM binding pose, the alpha-linkage is not recognized. Instead, the reducing end glucosyl unit stacks against Trp342 while the non-reducing unit extends into the catalytic pocket. Hence, a similar binding pose is conceivable for cellobiose and potentially also for chitobiose. The relevant section has been reworded.